# The Dilemma of Rapid AI Advancements: Striking a Balance between Innovation and Regulation by Pursuing Risk-Aware Value Creation

Lorenzo Ricciardi Celsi

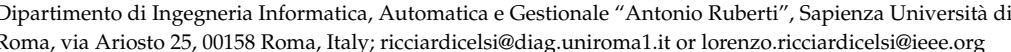

Dipartimento di Ingegneria Informatica, Automatica e Gestionale "Antonio Ruberti", Sapienza Università di Roma, via Ariosto 25, 00158 Roma, Italy; ricciardicelsi@diag.uniroma1.it or lorenzo.ricciardicelsi@ieee.org

**Abstract:** This paper proposes the concept of risk-aware actual value as a pivotal metric for evaluating the viability and desirability of AI projects and services in accordance with the AI Act. The framework establishes a direct correlation between the level of risk associated with a product or service and the resulting actual value generated. The AI Act reflects a concerted effort to harness the potential of AI while mitigating risks. The risk-based approach aligns regulatory measures with the specific attributes and potential hazards of distinct AI applications. As trilogue negotiations continue, the regulatory approach of the EU is evolving, highlighting its commitment to responsible and forward-thinking AI governance. Through a dedicated analysis of the AI Act, it becomes evident that products or services categorized as high-risk carry substantial compliance obligations, consequently diminishing their potential value. This underscores the imperative of exercising caution when engaging in projects with elevated risk profiles. Conversely, products or services characterized by lower risk levels are poised to accrue more substantial benefits from their AI and data potential, highlighting the incentive for a discerning approach to risk assessment. Methodologically, we propose an extension of an integrated AI risk management framework that is already existing in the literature, combining it with existing frameworks for measuring value creation from harnessing AI potential. Additionally, we contribute to the applied field of AI by implementing the proposed risk framework across nine industry-relevant use cases. In summation, this paper furnishes a comprehensive approach to achieving equilibrium between innovation and regulation in the realm of AI projects and services. By employing the risk-aware actual value metric, stakeholders are empowered to make informed decisions that prioritize safety and maximize the potential benefits of AI initiatives. This framework may stand as a reference point in this time when fostering responsible and sustainable AI development within the industry becomes of paramount importance.

**Keywords:** AI Act; AI regulation; AI risk

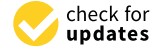



## 1. Introduction

Artificial intelligence (AI), by leveraging the three pillars of data, computational capabilities, and learning algorithms, is the discipline that reads past data to suggest future data, in order to make reliable predictions. With the advent of generative AI, it has also become capable of creating contents such as novel texts, codes, audio, images, digital art, and videos based on text prompts. While the classical computing paradigm follows rules, so classical programming does not scare us because it is predictable, AI has introduced a paradigm shift: we no longer design programs by telling the machine what to do in the form of a list of precise instructions. Instead, we instruct the machine to read the data and extract a suitable model from such data so that, based on the obtained model, the machine can return us its own decision rules. For example, until a few years ago, automatically translating a text was very difficult because classical programming was used. With AI, instead, the more data you feed the machine with, the better it will infer the next word for the text you need to be automatically translated. This paradigm shift generates scary

feelings in the end user—which are being mistakenly emphasized by mass media—because we only control which data we have given to the machine, but, since the machine makes correlations, we cannot know in advance which prediction model the machine is going to produce. More precisely, the machine only makes correlations: it cannot determine if there is a causal relationship between the correlated data [1]. All in all, the introduction of ChatGPT and the subsequent discourse highlight the pressing need for thoughtful deliberation on the future of AI.

In light of this, it is important to set up "guardrails" so that the machine's output falls within two limits: regulatory limits, on the one hand, and ethical limits, on the other. The machine equipped with AI is definitely not conscious: it is only "intelligent" in the sense that it carries out complex operations by multiplying matrices. In this sense, it is not even intelligent: we would rather talk about "artificial unconsciousness" [2]. In other words, thus far a machine equipped with AI has produced a result that makes it only "appear" as conscious, even though it is not. Even if studies are ongoing to recreate emotions artificially, most researchers agree that a logical-mathematical model (that is, the essence of a machine) will never be able to represent an exact copy of the human mind with its own intrinsic complexity [3,4]. Others, however, believe that we should prepare for the age of artificial general intelligence (AGI), which will equip AI systems with flexible human-level intellect [5].

In general, AI, like electricity in the past, is certainly proving a particularly valuable resource, but also a source of conflict, both in terms of the supply of advanced technologies and of the advantage that its possession can provide over the adversary, not just on the battlefield. This is why AI is playing a major role in today's geopolitical scenarios, and it is having enormous impact on the economic scenario, which in itself is a source of imbalances and tensions. One of the most pressing concerns is that of ensuring the trustworthiness of AI-generated content [6]. The provenance of information and its traceability are essential to maintaining the integrity of AI systems. Without addressing these issues, the proliferation of misinformation could erode the societal consensus on reality, leading to the manipulation of language, information, and knowledge. The influence that AI can have on public opinion, for instance by creating an enemy, is immense. Furthermore, the swift progression of AI technologies, exemplified by ChatGPT, has ignited debates on the appropriateness of its rapid deployment. The open letter from the Future of Life Institute [6], signed by prominent technologists and evangelists including Nell Watson and Grady Booch, emphasizes the lack of transparency in AI models and the need for a pause in research [7]. However, this call for caution stands in contrast to the growing integration of AI in various sectors.

Therefore, consensus arises about the necessity for a general project for AI that is "friendly to humans", or rather algorethics as a "compass" for these technologies [8], "best practices" to be inspired by, for instance, the so-called laws of technoethics drawn out a few years ago [9] or regulations based on human rights. In particular, the approach of the European Union to AI regulation stands out when compared to global efforts. Although the United States initially adopted a lenient stance, calls for AI regulation have grown. Both China and the UK are developing their own regulatory principles. International organizations such as the OECD and UNESCO have released recommendations on AI ethics. Additionally, the EU–US tech partnership is actively working towards establishing a common understanding of responsible AI principles. These collective efforts position the EU as a global frontrunner in AI governance. Meanwhile, corporations in the EU (e.g., Assicurazioni Generali [10]) are completely reviewing their vision and their values, so that their business units are strategically steered towards creating tangible business benefits that incorporate AI in all their business processes.

Definitely, different approaches are being explored, and ultimately a combination of these hypotheses will be reached. The interest in reaching this goal is also shared by the major religions, which have identified with the Rome Call for AI Ethics initiated by the Pontifical Academy for Life [11]. It is necessary to establish systems that are not competitive with respect to humans but complementary, contributing to the full realization of humans

without forming a sort of new species of sapiens. Then there is the idea of creating systems that do not exclude or marginalize the poorest, thereby avoiding the creation of new inequalities that could be at the root of possible conflicts [8].

Last but not least, concern is increasingly being raised about the dilemma of whether the relationship between humans and machines will change the labor market. Indeed, scholars agree that the labor market is not to be considered a zero-sum game: machines are not going to replace humans, especially as only humans are currently capable of creating relationships. If machines can replace us in operational tasks, this will definitely bring more benefits than costs because it will increase our time to be dedicated to relationships and those creative activities where pure machine calculation is not enough. Yet, in order to welcome this transformation comfortably, we need fast cultural, social, and political innovation to keep up with rapid technological innovation. The organizational culture of companies must change to embrace new, more efficient process models where AI tools are used more virtuously [12].

The European Union's proposal of an AI Act [13] offers a comprehensive approach to regulating AI technologies. Categorizing AI applications into risk levels, the Act outlines bans, stringent regulations, and limited regulation for high-risk, medium-risk, and low-risk applications, respectively. However, the Act still does not explicitly cover generative AI, which necessitates further consideration.

The tension between rapid innovation and effective regulation is a core challenge in AI integration. There is clearly a responsibility for tech CEOs, government bodies, and society at large in collaboratively shaping regulations that balance the benefits of AI with ethical considerations.

Indeed, striking a balance between AI innovation and responsible regulation remains a paramount challenge. As society grapples with the integration of AI technologies, it is imperative that individuals, institutions, governments, and corporations collaborate to shape a future that harnesses the potential of AI while safeguarding ethical and societal values. The path forward will therefore be a collaborative endeavor. We underscore the shared responsibility of stakeholders in guiding the future of AI. Navigating the complex landscape of AI requires collective efforts to ensure a secure and prosperous coexistence of humans and machines.

The aim of this paper is to briefly review the AI Act framework and provide a systematic evaluation of the risks incurred by nine AI projects from different industries (namely, insurance, finance, energy, telecommunications, and transportation) according to the AI Act framework itself. Projects are ranked based on a risk-aware value creation indicator obtained combining the data benefit indicator introduced in ref. [14] and the so-called SAFE risk-measurement approach introduced in ref. [15].

In addition to ref. [15], some other recent works exhibiting similar approaches are ref. [16,17]. In ref. [16], Zhang et al. provide a comprehensive overview of a broad array of inherent risks that can arise in AI systems, grouping such risks into two categories: data-level risk (e.g., data bias, dataset shift, out-of-domain data, and adversarial attacks) and model-level risk (e.g., model bias, misspecification, and uncertainty). Moreover, they highlight the research needs that have to be addressed for developing a holistic framework for risk management with respect to such AI systems. Instead, in ref. [17], Floridi and Cowls identify an overarching framework consisting of five core principles for ethical AI. Four of them are core principles commonly used in bioethics: beneficence, non-maleficence, autonomy, and justice.

The research reported in this paper differs from such works in that it proposes an extension of the integrated AI risk management framework introduced in ref. [15] in compliance with the emerging AI regulations. We integrate it with the data benefit index introduced in ref. [14], to compute the actual value generated by an AI product or service, which can be considered as "risk-aware" as it is assessed in compliance with the AI Act (according to the mapping of its requirements into the SAFE framework). This definition of risk-aware actual value implies that the higher the risk, the lower the actual generated value.

For instance, according to the AI Act, a high-risk product or service will certainly incur a high number of obligations, making it more difficult to comply with the regulation. Thus, it is expected to generate a lower amount of risk-aware actual value than a limited-risk or low-risk product or service. The resulting metric is therefore meant as a comprehensive indicator that discourages from activating AI projects and services with unacceptable risk levels and rewards those AI projects and services that not only have a low risk value but also reap more benefits from their AI and data potential.

The paper is organized as follows. Section 2 reviews the AI Act framework. Section 3 discusses the proposed risk-aware value creation framework, based on specific criteria for quantifying any incurred data-driven benefit and measuring AI-driven risk. Section 4 discusses the findings by proposing the systematic evaluation of the risks incurred by the nine considered projects. Concluding remarks end the paper.

## 2. The AI Act Framework: Balancing Innovation and Ethical AI Governance

The progression of the EU towards AI regulation began with the introduction of soft-law measures, including ethical guidelines and policy recommendations.

On 17 July 2020, the High-Level Expert Group on Artificial Intelligence coordinated by the European Commission proposed a new tool aimed at assisting designers who aim to develop high-level AI prototypes. The name of this tool, the Assessment List for Trustworthy Artificial Intelligence (ALTAI) [18], clarifies its objective, which is to build "reliable artificial intelligence". ALTAI is most likely the first design tool that translates ethical principles related to AI into a checklist to be followed, which is a dynamic and accessible list of control checks with the purpose of protecting against risks similar to those mentioned above. It is, therefore, a procedure that allows for the assessment of the compatibility of an AI project with ethical principles.

The ALTAI checklist allows for a structured verification of any AI project initiative's adherence to seven important ethical principles following an initial assessment of the protection of fundamental rights, according to the following scheme: (0) respect for fundamental rights, (1) human agency and oversight, (2) system robustness and security, (3) privacy and data governance, (4) transparency, (5) diversity and non-discrimination, (6) societal and environmental well-being, (7) accountability.

Yet, over the last three years, the transformative potential of AI has been so impactful that it has led to continuously increasing concerns about its ethical and societal implications. The EU has responded by proposing the AI Act, a regulatory framework aimed at achieving a delicate equilibrium between AI innovation and ethical governance. The introduction of the AI Act signifies a pivotal shift towards a legislative approach that aims to foster ethical AI development and deployment. This approach underscores the commitment by the EU to striking a harmonious balance between encouraging innovation and safeguarding fundamental rights, reflecting what has been termed a "human-centric" approach.

The AI regulatory approach of the EU stands in contrast to other international initiatives, demonstrating a proactive effort to establish ethical AI governance. The proposed AI Act indicates a shift from soft-law measures—as it previously was with the ALTAI checklist—towards a comprehensive legislative approach. This transition exemplifies the EU's commitment to ensuring responsible AI development.

In this section, we carry out an in-depth analysis of the EU's regulatory approach to AI through the proposed AI Act [13]. The Act aims to establish a comprehensive framework governing the development, deployment, and use of AI systems within the EU. We outline the central provisions of the Act, including the categorization of AI systems based on a "risk-based approach", prohibitions on specific AI practices, and transparency obligations. This analysis underscores the EU's endeavor to strike a harmonious balance between fostering innovation and ensuring responsible AI deployment.

*Key Provisions of the AI Act*

The AI Act aims to establish a cohesive legal framework governing AI systems within the EU. A prominent feature of this framework is the introduction of a "risk-based approach" [19] which entails the categorization of AI systems into four distinct groups based on risk assessment: namely, unacceptable risk, high risk, limited risk, and low/minimal risk. This stratification allows for tailored regulations that address specific risk levels. This approach seeks to tailor regulations according to the four potential risk classes associated with different AI applications.

- *Unacceptable Risk*. The AI Act explicitly prohibits AI practices that are deemed to present unacceptable risks to safety, rights, and societal well-being. Such practices include the utilization of manipulative subliminal techniques, exploitation of vulnerable groups, facilitation of social scoring by public authorities, and enabling real-time remote biometric identification for law-enforcement purposes. Some examples are the use of predictive justice in criminal sentencing in the presence of machine bias [20], social credit systems [21], and armed robots or robotic soldiers.
- *High-Risk*. While high-risk AI systems are not forbidden, they are subject to rigorous requirements and obligations. These requirements are designed to ensure the safety and adherence to fundamental rights of AI systems deployed in critical domains such as healthcare and transportation. Title III (Article 6) of the proposed AI Act regulates high-risk AI systems that could potentially compromise safety or fundamental rights. These systems are classified into two categories:
  1. Systems serving as safety components of products or falling under EU health and safety harmonization legislation, such as toys, aviation, and medical devices;
  2. Systems deployed in eight specific areas, as outlined in Annex III of the AI Act. The Commission can update this list through delegated acts, encompassing domains like biometric identification, critical infrastructure management, education, employment, and law enforcement.
- *Limited Risk*. These are AI systems that pose limited risks and are subject to minimal transparency obligations. Such AI systems presenting a "limited risk" include chatbots, emotion recognition systems, and image/audio/video manipulation AI, which are subject to modest transparency obligations. This approach acknowledges that not all AI applications warrant the same degree of regulation while still maintaining transparency where appropriate.
- *Low or Minimal Risk and Voluntary Obligations*. AI systems with a low or minimal risk would face no additional obligations. Nonetheless, the AI Act encourages voluntary adherence to mandatory high-risk requirements. This approach seeks to balance innovation and regulatory concerns.

A particular focus is made by the AI Act on facial recognition technologies (FRTs). Namely, the AI Act addresses FRTs, differentiating between "high-risk" and "low-risk" usage. Real-time FRTs for law enforcement are prohibited in publicly accessible spaces, except under specific circumstances. Other FRTs may be permitted if they adhere to safety requirements and undergo a conformity assessment.

EU lawmakers have already suggested amendments to the AI Act to enhance its effectiveness. The ongoing trilogue negotiations between the European Parliament, Council, and Commission aim to fine-tune the framework. These negotiations underscore the EU's commitment to addressing stakeholder feedback and enhancing the regulatory framework to ensure its efficacy. These proposed changes include revisions to the definitions of AI systems, an expanded list of prohibited AI practices, and the imposition of obligations on general-purpose AI and generative AI models.

Member states would appoint competent authorities, including a national supervisory body and a European Artificial Intelligence Board, to oversee the Act's application. Market surveillance authorities would assess high-risk AI compliance, including access to confidential data. Corrective measures and fines would be imposed for non-compliance,

underscoring the Act's enforcement provisions. To promote innovation, the Commission proposes regulatory sandboxes allowing controlled AI system development and testing before market introduction. This aligns with GDPR requirements and accommodates personal data usage for AI innovation. Additionally, measures cater to small-scale providers and start-ups.

One particular aspect is that distinct stakeholders differ on AI system definitions. Industry voices stress the broadness of the AI definition, while civil rights organizations advocate for an inclusive scope, encompassing systems in high-risk domains. The American Chamber of Commerce suggests narrowing the definition to focus on high-risk applications only.

Moreover, while stakeholders appreciate the risk-based approach, discussions include calls for broader prohibitions and regulations. Civil rights organizations demand limitations on AI use, especially for biometrics and predictive policing. Transparent impact assessments and clearer prohibitions are advocated for.

Indeed, there are a variety of perspectives on the EU's AI Act which reflect the complex nature of AI regulation. Stakeholders' insights on definitions, risk assessment, governance, enforcement, and innovation are critical in shaping effective AI governance. As legislative processes unfold, striking a balance between regulation and innovation remains paramount for the EU's AI regulatory framework.

## 3. Methodology

By combining the data benefit index (DBI) framework introduced in ref. [14] with the recent SAFE framework for risk measurement [15], this paper proposes a qualitative, explorative, and inductive methodology to investigate value creation from AI while preserving compliance with the guidelines expressed by the AI Act. An explorative qualitative methodology [22] comparing multiple projects and trying to generate meaning from the dataset collected in order to identify generalizing patterns and relationships seems to be the best option for developing empirically grounded hypotheses on emerging patterns on the use of AI.

The DBI framework is a useful tool which specifically allows us to compute the actual capacity of a data architecture to harness the data and AI potential within the project context. The DBI is computed as follows:

$$DBI = \frac{data\ consumption \times business\ value}{effort},$$

where the *data consumption* variable accounts for the number of users, the data literacy, and the time to market of the project outcome, the *business value* variable accounts for the value generated by the proposed AI and data solutions, evaluated by income growth, effectiveness at risk management, and operational efficiency, and eventually the *effort* variable accounts for the work required to implement a specific architectural approach, evaluated by operational expenditure, technical debt management, and data governance overhead.

The final DBI value is evaluated after normalizing the product *data consumption* × *business value* between 0 and 1 as well as the *effort* at the denominator of the ratio between 0 and 1. Each of these variables is quantified by assigning a score of 1 (low), 2 (medium), or 3 (high) that comprehensively considers the drivers listed above. The final score of a component is obtained by averaging the scores of the drivers that compose the index.

Depending on the resulting DBI value, we can distinguish among three different situations:

- $DBI < 0.5$: the data and AI potential are exploited to a limited extent;
- $0.5 \leq DBI < 1$: the data and AI potential are moderately exploited;
- $DBI \geq 1$: the data and AI potential are fully exploited.

Generally speaking, when DBI is greater than or equal to 1, the projects are making the most of the potential of data and AI for the reference industry, while projects whose DBI is between 0.5 and 1 are partially exploiting the potential that data and AI can express

for the reference industry. Finally, projects with a DBI lesser than 0.5 are not sufficiently exploiting the potential that data and AI can express for the reference industry.

Instead, to the best of the author's knowledge, the SAFE framework introduced in ref. [15] is the first risk-management framework for AI systems based on the recent regulation proposed by the AI Act. Namely, it maps the recently proposed regulatory requirements into a set of four measurable principles (sustainability, accuracy, fairness, explainability). Eventually, the four measurements are combined into a key AI risk indicator that allows us to classify the risk level incurred by the considered AI product or service.

More in detail, since the predictions from an AI model may be altered by the presence of "extreme" data points (either deriving from anomalous events or from cyber data manipulation), the driver of sustainability, denoted with $S$, accounts for the stability exhibited by the AI model against extreme variations in the input data.

The accuracy of an AI model, denoted with $A$, is the indicator with which it is possible to measure how often the AI model makes its decision correctly. More precisely, the accuracy is the number of correctly predicted data points out of all the data points. More formally, it is defined as the number of true positives and true negatives divided by the number of true positives, true negatives, false positives, and false negatives.

Fairness, denoted with $F$, is the property according to which an AI model does not present any biases among different population groups. Fairness should be verified at both the data (input) level and the prediction (output) level.

Instead, the explainability driver, denoted with $E$, accounts for the interpretability of the decisions made by the considered AI model: the more explainable the model is, the easier it is for auditors and regulators to validate AI models. Also, end-customers would like to be informed about the reasons for the predictions that involve them (e.g., in credit-lending applications) and on how their data are processed. In this respect, it is important to note that some AI models are explainable by design, in terms of their parameters (e.g., regression models), whereas others are black boxes (e.g., neural networks and random forests) which need a dedicated explainability layer in order to be suitably interpreted.

For the sake of clarity and simplicity, each of these variables is quantified by assigning a score of 1 (low), 2 (medium), or 3 (high), and the *Key AI Risk Indicator* ($KAIRI$) is computed as follows:

$$KAIRI = arithmetic\ mean(S, A, F, E)|_{[0,1]},$$

where the bracket operator $\cdot|_{[0,1]}$ accounts for the normalization of its argument between 0 and 1.

Depending on the resulting KAIRI value, we choose to distinguish among four different situations:

- $KAIRI \geq 0.9$: low or minimal risk;
- $0.7 \leq KAIRI < 0.9$: limited risk;
- $0.3 \leq KAIRI < 0.7$: high risk;
- $0 \leq KAIRI < 0.3$: unacceptable risk.

Finally, denoting it with $RAAV$, we now define the actual value generated by an AI product or service, which can be considered as "risk-aware" as it is assessed in compliance with the AI Act (according to the mapping of its requirements into the SAFE framework), as follows:

$$RAAV = KAIRI \times DBI.$$

This definition of risk-aware actual value implies that the higher the risk, the lower the actual generated value. For instance, according to the AI Act, a high-risk product or service will certainly incur a high number of obligations, making it more difficult to comply with the regulation. Thus, it is expected to generate a lower amount of risk-aware actual value than a limited-risk or low-risk product or service.

The risk-aware actual value is therefore meant as a comprehensive indicator that discourages from activating AI projects and services with an unacceptable risk level and rewards

those AI projects and services that not only have a low risk value but also reap more benefits from their AI and data potential. A flowchart describing how the DBI framework and the SAFE framework fit together for the computation of the RAAV is provided in Figure 1.

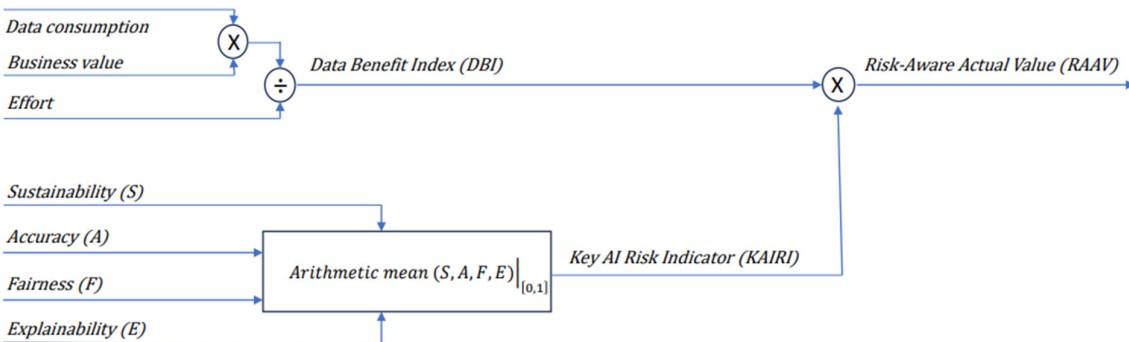

**Figure 1.** Flowchart describing how the DBI framework and the SAFE framework fit together to return the RAAV.

In the next section, we provide a comparative study among the different RAAV, DBI, and KAIRI values scored by nine different projects in different contexts in order to measure the level of risk with which the AI potential is being exploited in the related industries (namely, insurance, finance, energy, telecommunications, and transportation).

## 4. Findings

According to the DBI and SAFE frameworks, a previous screening work was conducted both in the literature and in the market in order to choose the projects that constitute our units of analysis. The criteria of choice, according to [23,24], were:

- presence of the AI-driven digital transformation as a primary aspect of the project;
- field of application;
- public evidence of the project.

To positively answer these criteria, we addressed our attention to nine AI data-driven projects, all of them published in international top journals.

The considered projects were carried out by either research groups or leading companies in the IT Consulting sector following the strategy discussed in Section 3. In particular, the vertical application scenarios addressed in this paper were recommended by the companies sponsoring the 2023 States General of AI, an event held in Italy on 28 February–2 March 2023 [25], according to the above-mentioned criteria of choice.

Below, we summarize the main characteristics of nine different projects that apply an AI-driven strategy. According to the DBI framework, the results of each project are organized on the basis of data consumption, business value, and effort. According to the SAFE framework, the risks of each project are organized on the basis of sustainability, accuracy, fairness, and explainability.

*Project #1—Convolutional neural networks for vehicle damage detection [26]*

This project proposes a damage-detection model developed to locate vehicle damages and classify these into twelve categories. Multiple deep-learning algorithms are used, and the effect of different transfer learning and training strategies is evaluated, to optimize the detection performance. The final model, trained on more than 10,000 damage images, is able to accurately detect small damages under various conditions such as water and dirt.

The consumption of AI and data solutions, based on the number of users, data literacy, and time-to-market, can be considered as medium-level (2) for this project. Indeed, the data literacy is not extremely high as the task is specifically focused on computer vision and this is still challenging when it comes to shifting from a working prototype to a profitable product. The business value too can be regarded as medium value (2), as the project outcome is not a game changer for the insurance industry but it would sensibly reduce

processing time from claim experts, thus improving task efficiency by about 50%. Yet the required effort is considerably high (3) by contrast with the other considered projects, especially in terms of opex (namely, the cloud infrastructure requirements for operation) and AI and data governance.

As per the risk-measurement framework, the sustainability variable can be regarded as low-level (1) because the main weakness of the underlying AI algorithms carrying out damage detection is that they are not stable enough against extreme variations in the input data, even though the training set dimensionality is quite high—this happens especially when the characteristics of the input data vary from the setting where the training images are taken. The accuracy variable in this case can be regarded as medium-level (2) because the proposed algorithms succeed in the antifraud task via computer vision with a mean average precision performance that is in line with the state of the art, yet there is still room for improvement—to be filled with more innovative computer vision architectures—in order to increase accuracy above 90%. While sustainability and accuracy can still be improved, fairness is high (3) regardless, as no particular bias among different population groups was detected through the extensive validation and testing campaigns conducted by the authors. Eventually, the explainability variable is to be assigned a low value (1) because the proposed computer vision architecture, being a deep neural network, is a black box and, to the best of the author's knowledge, the research community still has not investigated the predictors that best explain the decision made by AI in this use case.

*Project #2—A behavioral economics approach to predicting savings adequacy using machine learning [27]*

This project has the aim of proposing a machine-learning-based method that can predict individuals' savings adequacy in the presence of mental accounting. The proposed predictive model classifies wealth and consumption. Savings adequacy is best predicted by a decision-tree model. Surprisingly, it was found that future income and necessities have a lower predictive power on savings adequacy. The findings suggest that individuals, financial professionals, and policymakers should be cognizant that a higher likelihood of achieving savings adequacy can be achieved by focusing on the accumulation of current assets while lowering expenditure on luxury items.

In this case, the consumption of AI and data solutions, especially relative to the number and type of users of such a tool, as well as data literacy, can be regarded as relatively high: this is why we eventually choose to assign a high-level score to the data-consumption variable (3). The business value can be regarded as medium-level (2) too, as it lies in the marketing actions that can be targeted to each cluster of individuals as a result of segmentation. The effort, instead, is quite low (1), as operational expenditure, technical debt, and data governance are not a cause for concern: overall, it is an easy task to carry out provided that reliable data are used.

As per the risk-measurement framework, the sustainability variable can be regarded as high-level (3) in this use case because the proposed solution, being validated on a very large dataset accounting for almost 20 million individual savers, proves stable against extreme variations in the input data. The accuracy variable can be assigned a medium value (2) as the outcomes of data clustering are reliable enough, yet there is still room for improvement with the help of more advanced machine-learning techniques. The fairness can be assessed as high (3), as no particular bias among different population groups was detected through the extensive validation and testing campaigns conducted by the authors. The explainability, too, can be assessed as high (3), since the chosen methodology allows for a systematic interpretability analysis.

*Project #3—HR analytics for skill extraction using document embedding [28]*

This project was carried out to develop a job recommender system aimed at matching resumes to job descriptions, both of which are non-standard and unstructured/semi-structured in form. Namely, it proposes a combination of natural language processing (NLP) techniques for the task of skill extraction. Such a solution is expected to yield the following benefits for the Human Resources Department of a large company:

- Avoiding the manual and very often subjective collection of employees' training needs and minimizing the manual and subjective calculation of the return on investment of training, ultimately offering a choice of training courses that is necessarily in line with the employee's job profile;
- Optimization and adaptation of resource staffing on activities: by systematizing the entire professional and extra-professional career and aspirations of each employee, there is the possibility of more dynamic, efficient, and rigorous construction of work teams, saving time and making the staffing process more effective.

In this respect, data consumption and business value are both medium-level (2) but certainly the effort is quite high (3) as extensive NLP is needed for this task and this is critical in terms of operational expenditure and data governance.

As per the risk-measurement framework, the sustainability variable can be regarded as medium-level (2) in this use case because the proposed solution, in the absence of any standard large open-source dataset for the job-recommendation task, was trained and validated on a dataset consisting of resumes in Europass format and enriched with resumes created from as many LinkedIn profiles through the LinkedIn Resume Builder. The dimensionality of this dataset is not small, yet it still does not allow us to prove the algorithm stability against extreme variations in the input data. The accuracy variable can be assigned a medium value (2), as the accuracy performance at the first job recommendation is close to 90% but still a bit lower (namely, 89%). The fairness can be assessed as medium-level (2) because, even though no particular bias among different population groups was detected through the testing campaign carried out, a more extensive validation should still be made before clearing out any doubts about the algorithm fairness. The explainability can be assessed as low-level (1), since the proposed algorithm is to be considered as a black box model in the sense that it is not easy to interpret its prediction results and in this respect a suitable explainability layer should be added.

*Project #4—A multi-variable dynamic thermal rating (DTR) algorithm for the estimation of conductor temperature and ampacity on high voltage overhead lines by IoT data sensors [29]*

This project had the aim of proposing a dynamic thermo-mechanical model approach for estimating the conductor's temperature and ampacities of power grids based on the weather data measured by IoT sensors.

In this case, the data consumption can be regarded as relatively low (1) in comparison with the other considered project, while the implied business value is certainly medium-level (2): indeed, dynamic thermal rating systems and the georeferencing of the electrical system represent an important evolutive step of a high-voltage network towards an intelligent cyber-physical system, and, through the possibility to continuously monitor several fundamental parameters related to the system (such as the temperature and the voltage of the conductors), they enable a more flexible operation of the rating of the overhead power lines, estimating temperature and ampacity with high reliability. The required effort is quite high (3), in contrast, as some parts of the Italian power grid are covered by the necessary infrastructure for data collection, but others still are not.

As per the risk-measurement framework, to guarantee the sustainability of the AI model it is necessary to carry out an extensive validation by installing direct temperature sensors throughout the whole power transmission network and comparing their measurements with the outputs of the AI model. This is why we choose to assign a low value (1) to the sustainability variable. The accuracy variable can be assigned a medium value (2) as there is still room for improvement in order to increase the estimation accuracy above 95% with the help of more advanced techniques. The fairness can be assessed as high (3), as no particular bias among different population groups is expected with the current setting. Finally, the explainability can be assessed as medium-level (2) as the nature of the proposed model (combining a thermal model with a mechanical model and a Bayesian approach) already gives a good degree of interpretability of the root causes of the predictions made; yet, augmenting the proposed model with the data related to the conductor distance from

the ground would increase such a degree of interpretability, while providing an even more precise evaluation of the current ampacity.

*Project #5—Anomaly detection and predictive maintenance for photovoltaic systems [30]*

This project presents a learning approach designed to detect possible anomalies in photovoltaic systems in order to let an operator to plan predictive maintenance interventions. By running the proposed algorithm on unseen data streams, it is possible to isolate anomalous conditions and virtually trigger an alarm when exceeding a reference threshold. The proposed approach was tested on both standard operating conditions and an anomalous scenario. With respect to the considered use case, it successfully anticipated a fault in the equipment, but also demonstrated its robustness to false alarms during normal conditions. Such a solution is expected to yield an overall downtime reduction between 1% and 2%, which corresponds to an increase in the annual photovoltaic panel production in the order of approximately 1–2 megawatts.

In this respect, data consumption can be regarded as medium-level (2), as it is necessary to collect and prepare data from selected sensors mounted onto the production line stations which are in charge of measuring several relevant parameters characterizing each station, such as the station temperature, pump speed, flow speed, and ozone concentration level. On the other hand, the associated business value is high-level (3), as the proposed solution implies a relevant benefit in terms of increase in the annual photovoltaic panel production and paves the way for the creation of a digital twin of the whole solar cell production plant. By contrast, the required effort is medium-level (2), especially in terms of opex and data governance.

As per the risk-measurement framework, the algorithm was primarily designed to ensure robustness with respect to false alarms, which implies a high sustainability score (3). The prediction accuracy is good but still leaves room for further improvement, which implies a medium-level score (2). The fairness can be considered as a high value (3), since no bias among different population groups was detected. The explainability, too, scores a high value (3) because the algorithm is also capable of identifying the very precursors of a failure event, thus allowing us to interpret the final decision of detecting the failure itself.

*Project #6—Explainable AI for car crash detection using multivariate time series [31]*

This project has the aim of explaining the decisions made by an AI model employed to optimize the assistance service of the insurance company. Whenever the vehicle is involved in an accident, the AI agent processes the deceleration burst collected by the black box onboard and automatically triggers a call to an insurance operator who, in turn, dials the driver to check his/her conditions and the general situation. If necessary, this call is forwarded to the emergency medical services, to offer any necessary pre-hospital treatment, and/or to the tow truck in charge of removing the vehicle itself.

The data consumption in this respect can be regarded as low (1), as the multivariate time series collected by the black boxes onboard vehicles are relatively easy to gather and process. The resulting business value is definitely high (3), as the optimization via the explainable AI of the automatic assistance service is expected to considerably increase the accuracy and trustworthiness of the predictions, thus allowing the company to bring to market a robust product. On the other hand, the effort is as very high (3), especially in terms of opex and AI governance.

As per the risk-measurement framework, since the considered use case relies on a neural network architecture to address the problem of classifying crash against non-crash events based on telematics data coming from black boxes mounted onboard insured vehicles, the sustainability variable deserves a medium score (2), as there exist in the literature classifiers which proved to be more robust in the presence of extreme variations of the input data. Similarly accuracy deserves a medium score (2), as it remains slightly below 90% and fairness (2), as no particular bias among different population groups was detected with respect to a dataset containing almost 900 crash events and 20,000 non-crash events, but the authors did not verify whether such fairness property is kept with an even larger reference database. Finally, the explainability variable scores the maximum value (3),

as a dedicated explainability layer was introduced in order to successfully interpret the black box model decision, comparing the performance of three different explainability methods (namely, Integrated Gradients, Grad-Cam and LIME).

*Project #7—Online service function chain deployment for live-streaming in virtualized content delivery networks (CDNs) via deep reinforcement learning (DRL)* [32]

This project has the aim of exploiting 5G networks to enable higher server consolidation and deployment flexibility in video delivery services. Namely, the proposed approach consists of a multi-objective optimization framework for service function chain deployment in the particular context of live-streaming in virtualized content delivery networks via deep reinforcement learning.

In this case, the amount of actual data consumption can be regarded as relatively low (1) if compared with the other projects. Yet, the business value and effort are both quite high (3), the former due to the considerable benefits yielded by the proposed solution in terms of quality of service and quality of experience, whereas the latter is due to the burst in operational expenditure and technical debt that results from investing in such an activity.

As per the risk-management framework, the sustainability variable deserves a low score (1), as the experiments are based on a real-world dataset concerning a particular video delivery operator and limited to a five-day trace, so further research should assess the agent's training and evaluation performance on data concerning more extended periods, before going into production. The accuracy in this case is medium-level (2), as all key performance indicators (scaled throughput, acceptance ratio, rewards, scaled costs, scaled hosting costs, and objective function) for the proposed enhanced-exploration dense-reward dueling double deep Q-learning (E2-D4QN) algorithm are encouraging but still a bit lower than top quality values. The fairness is medium-level (2) too, as no particular bias was detected but the possible population groups were extracted out of a dataset accounting for relatively small amount of time. The explainability can be assessed as low-level (1), since the proposed E2-D4QN engine is to be considered as a black box model in the sense that it is not easy to interpret its prediction results and in this respect a suitable explainability layer should be added.

*Project #8—Ticket automation* [33]

This project deals with the problem of handling large amounts of customer requests: machine-learning algorithms are fundamental in streamlining support ticket processing workflows. Two commonly used datasets, both characterized by a two-level hierarchy of labels, are employed and they are descriptive of the ticket's topic at different levels of granularity. The first is a collection of 20,000 customer complaints, and the second comprises 35,000 issues crawled from a bug reporting website. Using these data, the project focuses on topically classifying tickets using a pre-trained BERT language model.

In this respect, the amount of data consumption is to be regarded as quite high (3), especially as regards the number of users involved. By contrast, the business value can be considered as medium-level (2), since the proposed solution proves to be a very useful tool for service providers in order to identify the customers that are most at risk of reopening a ticket due to an unsolved technical issue. The required effort, too, is medium-level (2), especially in terms of operational expenditure.

As per the risk-measurement framework, the proposed approach was validated on an extremely large dataset considering the problem at hand, so we can assign a high sustainability score (3), as the considered dataset includes enough variations to ensure the robustness of the adopted algorithm. The accuracy can be given a medium-level score (2), as it achieves a good result (up to 80%) but still not an excellent one (which should be above 90%). The fairness achieves a high score (3), as no particular bias was detected among the different population groups extracted out of the above-mentioned very large validation dataset. Finally, the explainability variable deserves a medium-level score (2), as the proposed algorithm certainly unveils the root causes of the ticket reopening phenomenon outlining specific predictors; yet, there is still room for improvement in order to better

explain the ultimate prediction of whether a monitored ticket will be reopened or not. This improvement could be achieved, as in other considered use cases, with a dedicated explainability layer.

*Project #9—Enhancing traveler experience in integrated mobility services via big social data analytics [34]*

This project had the aim of proposing a data-driven approach to boost the tourist experience in integrated mobility services: in particular, owing to the design of a recommendation system based on a big-data analytics engine, it makes it possible to rank the tourist preferences for the most attractive Italian destinations on Google, and to rank the main attractions—leisure, entertainment, culture, etc.—associated with single tourist destinations, obtained from the analysis of relevant thematic websites such as TripAdvisor, Minube, and Travel365.

In this case, the data consumption can surely be evaluated as low (1), together with the required effort mostly in terms of data governance (1), especially by comparison with the other mentioned projects. In contrast, the business value is medium-level (2), as such a solution allows mobility companies to define targeted and more effective marketing campaigns via a relatively easy collection of data from several heterogeneous sources such as Google search queries accessible via Google Trends, or any social data scraped from websites.

As per the risk-measurement framework, the proposed recommender system was trained and validated on a very large dataset obtained by crawling on three travel social networks, namely TripAdvisor, Minube, and Travel365, so we can assign in this use case a high sustainability score (3), as the considered dataset includes enough variations to ensure the robustness of the adopted algorithm. The accuracy can be given a medium-level score (2), as it achieves a good result (up to 80%) but still not an excellent one (which should be above 90%). The fairness achieves a high score (3), as no particular bias was detected among different population groups extracted out of the above-mentioned very large set of big social data. Finally, the explainability variable deserves a medium-level score (2), since the proposed recommender system bases its predictions on a suitably arranged popularity index, which, despite having a relatively satisfactory interpretability property with respect to the final decision made, would certainly benefit from the introduction of a dedicated explainability layer (yet to be covered by the authors).

As is clear from the project descriptions, creating value from data and AI is a complex task, especially due to the difficulty in aligning business objectives with the AI-driven digital transformation, while keeping the related risks limited. The RAAV framework captures this need: based on that, we report in Table 1 the rankings of the above-introduced projects by decreasing DBI, and in Table 2 the rankings of the above-introduced projects by decreasing KAIRI. The results shown in Table 2 imply the risk classification given in Table 3, which reflects the KAIRI framework being compliant with the AI Act framework. Each project should therefore satisfy the requirements recommended by the AI Act for each risk category. Among the considered projects, none turned out to be of unacceptable risk, as none of them unacceptably threatens safety, rights, and societal well-being.

**Table 1.** Ranking of the nine projects evaluated by decreasing DBI.

| Project Name | *data consumption* | *business value* | *effort* | *DBI* |
|---|---|---|---|---|
| *#2—Predicting savings adequacy* | 3 | 2 | 1 | 2.28 |
| *#5—Anomaly detection and predictive maintenance for photovoltaic systems* | 2 | 3 | 2 | 1.50 |
| *#8—Ticket automation* | 3 | 2 | 2 | 1.50 |
| *#9—Big social data analytics for enhancing traveller experience* | 1 | 2 | 1 | 1.26 |
| *#6—Explainable AI for car crash detection* | 1 | 3 | 3 | 0.67 |

**Table 1.** *Cont.*

| Project Name | data consumption | business value | effort | DBI |
|---|---|---|---|---|
| #7—*Live streaming in virtual CDNs via DRL* | 1 | 3 | 3 | 0.67 |
| #1—*Convolution neural networks for vehicle damage detection* | 2 | 2 | 3 | 0.50 |
| #3—*HR analytics for skill extraction* | 2 | 2 | 3 | 0.50 |
| #4—*DTR in power transmission grids* | 1 | 2 | 3 | 0.42 |

**Table 2.** Ranking of the nine projects evaluated by decreasing KAIRI.

| Project Name | S | A | F | E | KAIRI |
|---|---|---|---|---|---|
| #2—*Predicting savings adequacy* | 3 | 2 | 3 | 3 | 0.92 |
| #8—*Ticket automation* | 3 | 2 | 3 | 2 | 0.83 |
| #9—*Big social data analytics for enhancing traveller experience* | 3 | 2 | 3 | 2 | 0.83 |
| #6—*Explainable AI for car crash detection* | 2 | 2 | 2 | 3 | 0.75 |
| #4—*DTR in power transmission grids* | 1 | 2 | 3 | 2 | 0.67 |
| #5—*Anomaly detection and predictive maintenance for photovoltaic systems* | 1 | 2 | 2 | 3 | 0.67 |
| #1—*Conv neural networks for vehicle damage detection* | 1 | 2 | 3 | 1 | 0.58 |
| #3—*HR analytics for skill extraction* | 2 | 2 | 2 | 1 | 0.58 |
| #7—*Live streaming in virtual CDNs via DRL* | 1 | 2 | 2 | 1 | 0.5 |

**Table 3.** Risk level classification of the nine projects according to the AI Act as a result of applying the combined DBI-KAIRI approach.

| Project Name | Risk Class |
|---|---|
| #1—*Conv neural networks for vehicle damage detection* | High |
| #2—*Predicting savings adequacy* | Low or minimal |
| #3—*HR analytics for skil extraction* | High |
| #4—*DTR in power transmission grids* | High |
| #5—*Anomaly detection and predictive maintenance for photovoltaic systems* | High |
| #6—*Explainable AI for car crash detection* | Limited |
| #7—*Live streaming in virtual CDNs via DRL* | High |
| #8—*Ticket automation* | Limited |
| #9—*Big social data analytics for enhancing traveller experience* | Limited |

*Implications*

Methodologically, above we proposed an extension of the integrated AI risk-management framework initially introduced by [15], combining it with existing frameworks for measuring value creation from harnessing AI potential. Additionally, we contribute to the applied field of AI by implementing the proposed risk framework across nine industry-relevant use cases. In this respect, Figure 2 reports a synoptic view of DBI, KAIRI, and RAVV values, reporting the nine projects by decreasing RAVV. From the combined analysis of DBI and KAIRI values given in Figure 2, it turns out that financial institutions (as regards Project 2) are the most mature and ready to harness the data and AI potential with low or minimal risk. Then, specific use cases in the energy, telecommunications, and transportation industries—namely, Project 8 with reference to the telecommunications scenario, and Project 9 with reference to the

transportation scenario—pose challenges which strike the right balance between advanced and effort-requiring techniques, on the one hand, and limited risk, on the other hand, whereas Projects 5 and 7, despite the high risk, are expected to bring a great deal of benefit (Project 5 even more than Project 7), so they are still a recommended investment provided that the necessary obligations descending from the AI Act are rightfully put into practice. The domain of insurtech—as per Projects 6 and 1 ordered by decreasing KAIRI—follows, together with specific use cases in the energy and telecommunications industries. In this last case, the implied AI risk rises significantly, suggesting to wait before investing into bringing such projects into production, especially as the challenge of deploying the adopted technologies (namely, NLP for Project 3 and DTR for Project 4) currently yields higher costs than benefits.

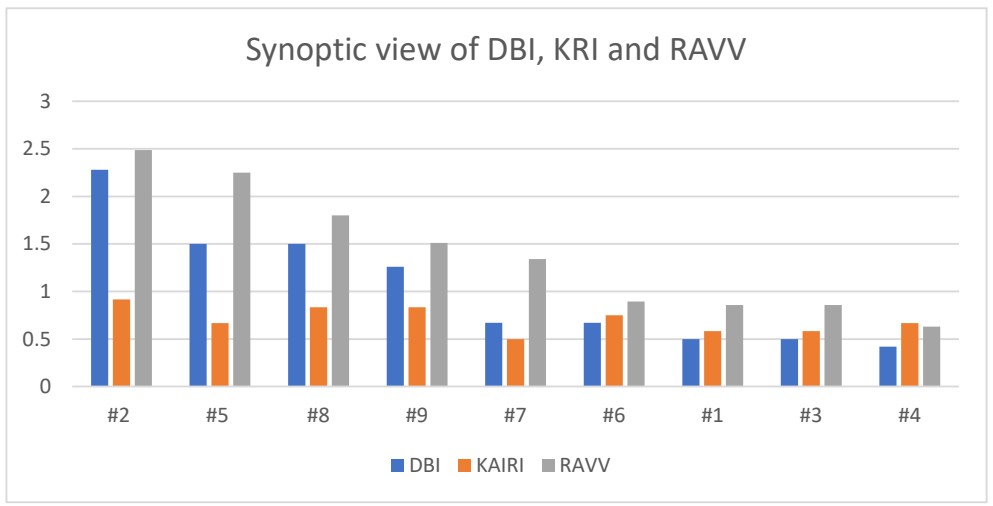

**Figure 2.** Synoptic view of DBI, KAIRI, and RAVV values. The nine projects are reported by decreasing RAVV.

By employing the risk-aware actual value metric, stakeholders are empowered to make informed decisions that prioritize safety and maximize the potential benefits of AI initiatives. This framework may stand as a reference point in this time when fostering responsible and sustainable AI development within the industry becomes of paramount importance.

The contribution the findings of the project analysis add to the existing knowledge about the ethical and legal use of AI lies in the proposition of a risk-aware actual value indicator for a project or investment in AI technology. This implies that the higher the risk, the lower the actual generated value. For instance, according to the AI Act, a high-risk product or service will certainly incur a high number of obligations, making it more difficult to comply with the regulation. Thus, it is expected to generate a lower amount of risk-aware actual value than a limited-risk or low-risk product or service. The risk-aware actual value is therefore meant as a comprehensive indicator that discourages from activating AI projects and services with an unacceptable risk level and rewards those AI projects and services that not only have a low risk value but also reap more benefits from their AI and data potential. This result may help improve existing regulations, especially in Europe, using a bottom-up or inductive approach: from the very results of existing AI projects, it is possible to infer relevant insights for improving the ethical regulation of AI design and use.

In more general terms, it is clear from the topics discussed above that there are a series of issues arising from the impact of AI in industry. The repercussions of AI on current project practice are to be assessed in terms of the following themes.

- *Data security and privacy concerns.* In particular, the storage and protection of data may raise concerns about the security and privacy of sensitive information, especially if the project involves handling large amounts of personal or confidential data. Also, the need for robust data protection may drive the adoption of privacy-preserving

techniques in AI, such as federated learning or homomorphic encryption, which allow for AI model training without exposing raw data.

- *Compliance with already existing regulation*. If projects reveal vulnerabilities in data protection that could lead to non-compliance with data protection regulations (e.g., GDPR, HIPAA), organizations may need to reassess their practices to avoid legal consequences.
- *Impact on trust and adoption*. Issues related to data security can erode trust in AI systems. If stakeholders, including customers, lose confidence in the ability of organizations to protect their data, this could slow down the adoption of AI technologies.
- *Increased emphasis on ethical AI.* The importance of ethical considerations in AI development is such that organizations are increasingly prioritizing ethical frameworks and guidelines to address potential biases, discrimination, or misuse of AI systems.
- *Investment in cybersecurity measures*. Organizations may need to allocate more resources to enhance cybersecurity measures, including encryption, access controls, and regular security audits, to safeguard the data used in AI projects. Organizations might also establish closer collaborations between AI experts and cybersecurity professionals to ensure a comprehensive approach to data protection in AI projects.
- *Impact on project planning and execution*. Project managers may need to incorporate robust data protection measures into the early stages of AI projects, affecting project timelines and resource allocation.
- *Educational initiatives*. Organizations increasingly feel the need to invest in educating their teams about the importance of data security in AI projects, ensuring that personnel are well-versed in best practices and potential risks.
- *Insurance and risk management*. With increased awareness of potential risks associated with AI projects, organizations may explore or expand insurance coverage related to data breaches and AI-related liabilities.

## 5. Conclusions

In this paper, after reviewing the risk-based framework proposed by the AI Act, we introduce the concept of risk-aware actual value which we propose as a pivotal metric in evaluating the viability and desirability of AI projects and services. Indeed, it establishes a clear correlation between the level of risk associated with a product or service and the resultant actual value generated.

As exemplified by the provisions outlined in the AI Act, products or services categorized as high-risk carry a substantial burden of compliance obligations, consequently diminishing their potential value. This underscores the imperative to exercise caution when engaging in projects with elevated risk profiles. Conversely, products or services characterized by lower risk levels stand to garner more substantial benefits from their AI and data potential, highlighting the incentive for a judicious approach to risk assessment.

By embracing the risk-aware actual value as a comprehensive indicator, stakeholders are empowered to make informed decisions that prioritize safety and maximize the potential benefits of AI initiatives. This framework, thus, can be considered as a reference point in fostering responsible and sustainable AI development in industry.

From a methodological viewpoint, we propose an extension of the integrated AI risk-management framework introduced in ref. [15] in compliance with the emerging AI regulations.

From an applied viewpoint, we contribute to the research on the application of AI by implementing the proposed risk framework for evaluating value creation in nine use cases, which have been indicated by the industry among the most relevant and necessary applications of AI.

The main difficulty encountered by the author lay not in applying the presented methodology or performing the analysis on the projects but in identifying a set of indicators that may be versatile enough to evaluate the data benefit index, the key AI risk indicator, and consequently the risk-aware actual value on any AI-driven project scenario. Thus, most of the research effort was devoted to defining the minimum set of variables—as reported in Figure 1—that concur to measuring the data benefit index, the key AI risk indicator,

and therefore the risk-aware actual value, while ensuring the applicability of the proposed framework to the widest range of industrial scenarios where an investment in AI may prove strategic.

Further research is required from a mathematical and statistical viewpoint, to propose alternative ways to measure the considered indicator and test its statistical significance, thereby providing tools that can improve the robustness of the conclusions drawn.

**Funding:** This research received no external funding.

**Data Availability Statement:** No new data were created.

**Conflicts of Interest:** The author declares no conflict of interest.

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
