# Peer review of "The Dilemma of Rapid AI Advancements: Striking a Balance between Innovation and Regulation by Pursuing Risk-Aware Value Creation"

_information, doi:10.3390/info14120645_

Round 1

Reviewer 1 Report

Comments and Suggestions for Authors

The paper addresses one of the most relevant yet less studied problem in AI: Balancing act  of  innovation versus ethical consideration. With the proliferation of AI supported systems it is imperative to critically evaluate the direct as well as implied impact of AI in the broader scope of human endeavours. The paper uses EU AI Act as a reference point to propose a system for evaluation of AI systems based on reward versus risk framework. The indicators for benefit and risk scores are normalized and combined to yield their proposed RAAV score. The proposed methodology would significantly benefit by incorporating relevant statistical techniques.

The paper is easy to read and the use-cases do provide evidence of applicability of the proposed method.

Specific (minor) comments:

Line 106: The sentence is unclear.

Line 184: Remove ``This''

Line 292: Please clarify the statement ``Instead, ..."

Line 301; ``estreme'' should be ``extreme''

Line 339: Something isn't right in the sentence.

Line 354: What is meant by ``Public evidence of the project"    

Author Response

The author thanks the reviewer for the positive evaluation of the manuscript. Detailed responses are specified in the attached document.

Reviewer 2 Report

Comments and Suggestions for Authors

-The paper proposes the concept of risk-aware actual value as a pivotal metric for evaluating the viability and desirability of AI projects and services according with the AI Act.

-Interesting work that could help in assessing AI innovation and its related regulation in terms of risk-aware value.

-The technical aspects of the research presented seem sound and detailed.

-While you mentioned it in the "Introduction" section, the contribution and potential practical applications of your work should be stated in a clearer way.

-Regarding the writing style and typos in the paper, I would suggest that the authors to double check the text as there are some typos in the paper.

-It would be useful for the reader if you include a diagram or system architecture showing the different stages of your proposed approach.

-I suggest to make changes in Figure 1 to improve readability. For example, the text below each set of bars could be simplified by indicating just the number of the project. 

Comments on the Quality of English Language

-Regarding the writing style and typos in the paper, I would suggest that the authors to double check the text as there are some typos in the paper.

Author Response

The author thanks the reviewer for the positive evaluation of the manuscript.

Detailed responses to the reviewer's comments can be found in the attached document.

Reviewer 3 Report

Comments and Suggestions for Authors

This manuscript deals with a hot topic in the artificial intelligence (AI) discussion both in the academy and in the industrial/market sides. Currently, with the ascension of large language models and generative artificial intelligence, several nations started a debate about the development of a regulatory/legal framework for the use of AI by society, especially in the case of the market/industry. The related scenario is analyzed in the European context; however, it represents a trend worldwide towards AI regulation.

The references cited in the manuscript are up-to-date, and the theoretical references are well-developed. However, I would like to see a list of related works. It is not necessary to be extensive in the number of works, but I think it is important to present some other recent works with similar approaches, highlighting how the research reported in this manuscript differs.

Below I indicate some elements to be improved in the work:

1.       The paper objective is clearly defined: review the AI Act framework providing an evaluation of risks related to a set of AI projects under the light of the framework. However, I missed a declaration of the research problem. Please provide your research problem. It can be presented in the form of a research question, for instance. If you have premises related to the problem, please, also state them.

2.       Please, in the methodology, provide a flowchart related to the research process applied. I understood your approach applied a combination of the DBI framework with the SAFE framework, but for me, it is not so clear how the two fit together. With a flowchart, the final composition of both frameworks can be better understood. Remember to link the flowchart to the current text in the methodology section.

3.       I also understood that the projects selected for analysis were chosen based on a screening of literature and the market. It used three criteria, as presented in the findings section between lines 352 and 354. Please comment on where these criteria came from. Did you find them in the literature? Were they recommended by companies, by market practitioners? It is important to make this clear in a paragraph following the presentation of these three criteria.

4.       On page 11, line 513 with an indication of Project #6, there is a link (the only one for a project). Is the intention really to leave the text linked? Please consult the guidelines for authors to verify that this is the correct form.

5.       I miss an implications section in your work. You presented information on the projects analyzed, however, there is no summary of the practical and theoretical implications that you obtained from the analysis of the projects with the addition of the frameworks that you used:

a.       From a theoretical point of view, try to answer questions such as: what contribution do the project analysis findings contribute to the existing knowledge about the ethical and legal use of AI? How can the results obtained help to improve existing regulations, especially in Europe? Try to try to elucidate how your research fills any gaps in knowledge that exist on your research topic.

b.       From a practical point of view, look for evidence of how your findings about projects can impact new projects using artificial intelligence. There are a whole series of issues involving, for example, the storage and protection of data to be used by the industry to enable the use of AI. What are the repercussions of this on current practice, given what you observed in the projects analyzed?

6.       In the conclusions, it is important to mention your research limitations and evident challenges. What were the main difficulties in applying the presented methodology, and performing the analyses of the projects?

Author Response

(The authors gave the same response as above.)

Round 2

Reviewer 3 Report

Comments and Suggestions for Authors

The authors made the adjustments they recommended and answered the questions they highlighted in the previous round.

I believe that this new version is sufficiently adjusted to proceed for final acceptance.

Congratulations to the authors for the competently developed study on the topic of AI regulation.